# Prediction of Time-Series Transcriptomic Gene Expression Based on Long Short-Term Memory with Empirical Mode Decomposition

**DOI:** 10.3390/ijms23147532

**Published:** 2022-07-07

**Authors:** Ying Zhou, Erteng Jia, Huajuan Shi, Zhiyu Liu, Yuqi Sheng, Min Pan, Jing Tu, Qinyu Ge, Zuhong Lu

**Affiliations:** 1State Key Laboratory of Bioelectronics, School of Biological Science & Medical Engineering, Southeast University, Nanjing 210096, China; habbyzy@sina.com (Y.Z.); jiaerteng213@163.com (E.J.); 18705168526@163.com (H.S.); 15195986641@163.com (Z.L.); sheng_yuqi@163.com (Y.S.); jtu@seu.edu.cn (J.T.); zhlu@seu.edu.cn (Z.L.); 2School of Medicine, Southeast University, Nanjing 210097, China; 101012091@seu.edu.cn

**Keywords:** long short-term memory, time-series, gene expression, empirical mode decomposition, intrinsic mode functions

## Abstract

RNA degradation can significantly affect the results of gene expression profiling, with subsequent analysis failing to faithfully represent the initial gene expression level. It is urgent to have an artificial intelligence approach to better utilize the limited data to obtain meaningful and reliable analysis results in the case of data with missing destination time. In this study, we propose a method based on the signal decomposition technique and deep learning, named Multi-LSTM. It is divided into two main modules: One decomposes the collected gene expression data by an empirical mode decomposition (EMD) algorithm to obtain a series of sub-modules with different frequencies to improve data stability and reduce modeling complexity. The other is based on long short-term memory (LSTM) as the core predictor, aiming to deeply explore the temporal nonlinear relationships embedded in the sub-modules. Finally, the prediction results of sub-modules are reconstructed to obtain the final prediction results of time-series transcriptomic gene expression. The results show that EMD can efficiently reduce the nonlinearity of the original data, which provides reliable theoretical support to reduce the complexity and improve the robustness of LSTM models. Overall, the decomposition-combination prediction framework can effectively predict gene expression levels at unknown time points.

## 1. Introduction

The fate of RNA transcripts in dead tissues and the decay of isolated RNA are not subject to strict regulation, unlike in vivo normal cells [1]. Whether most transcripts decay at a similar rate remains unknown at present [2]. Understanding RNA degradation patterns is critical for studies of samples collected in the field or samples collected in clinical settings, as these tissue samples are often not immediately stored under conditions that can prevent RNA degradation [3]. For example, obtaining donated samples in the clinic may require several hours of transport for RNA extraction and sequencing analysis. It is crucial to investigate the decay pattern of RNA degradation over time, because neglecting the initial RNA quality changes in some clinical samples may even lead to incorrect judgments of disease. In addition, some RNA extraction methods need to take several hours. Currently, samples collected during certain low-quality fieldwork remain the only means of solving specific problems [4]. In these cases, the extracted RNA is usually partially degraded and may not faithfully represent the initial gene expression level [5]. Moreover, ignoring the initial RNA quality changes in clinical samples may lead to misjudgment of the disease. Analyzing the effect of RNA degradation on gene expression is essential to obtain meaningful and reliable gene expression data and to ensure reproducible results. Therefore, an artificial intelligence method is needed to predict the decay pattern of RNA degradation over time. If the irreversible change in gene expression level of RNA that occurred during this time can be understood clearly, it will be possible to predict the true gene expression level of RNA when degradation has not occurred by the time of degradation and the existing gene expression level.

Time-series-based gene expression data have become one of the most fundamental methods for studying biological processes. Most biological processes are dynamic, and using time series data can characterize the function of specific genes, discover linkages and regulatory relationships between genes, and find their clinical applications [6,7,8]. Time series data can not only capture transient changes in gene expression, but also demonstrate the order of events that occur during biological processes and allow the study of the temporal patterns of gene expression [9]. These transcriptome time series experiments require observational sampling of biological systems, the preparation of transcriptome libraries, and sequencing at each time point. The current time series experiments have a high overhead. Their sampling time points are generally limited and they are only used with some specific study subjects. Artificial intelligence computational methods are particularly important for humans, which can only perform limited experiments directly. Therefore, there is an urgent need for tools that can study limited time-series-based gene expression data or frameworks and methods that can use time-series-based gene expression data for analytical modeling [10]. So far, this field has been developed and studied by lots of scientists. For example, Lakizadeh et al. [11] proposed the use of time-series-based gene expression data to complement protein–protein interaction (PPI) networks to detect protein complexes. Wise et al. [12] developed SMARTS, a method that combines static and time-series-based data from multiple individuals to reconstruct condition-specific unsupervised feedback networks. Qian et al. [13] used deep neural networks for time-series classification tasks. However, there is no method that combines long short-term memory (LSTM) [14] deep learning models for gene expression prediction of time series.

It is the first prediction framework constructed by deep learning methods for time series data of isolated RNA from tissue samples. The time-series-based prediction is able to extract causality because the information on gene expression flows over time in one direction. Overall, we developed a new LSTM-based deep learning model named Multi-LSTM to predict gene expression at target time points in transcriptome time series data. (Figure 1). This model is capable of predicting the gene expression at initial time points from the gene expression at subsequent time points. It can also predict the gene expression at future times from previous time points. This study has important implications for those studies that lack initial time samples. Moreover, applying the LSTM deep learning model to the analysis of transcriptome time series data can obtain the missing RNA-seq gene expression data, which can facilitate the functional prediction of transcriptome gene expression profiles and the discovery of disease-related genes. With the increase in the number of time-series-based biological experimental data, it will be possible to make full use of the time-series information based on this study to gain a deeper understanding of the gene functions and molecular mechanisms in biological processes.

## 2. Results and Discussions

### 2.1. Multi-Scale Analysis Based on EMD

Our data are the expressions of transcripts corresponding to each gene obtained by extracting RNA from mouse brain tissue after transcriptional library building and sequencing. Usually, the normalized Fragments Per Kilobase of exon model per Million mapped fragments (FPKM) value is a standard value to judge the high or low expression of that gene [15]. In this study, the initial gene expression values were normalized as input values and predicted values. Two main time points were predicted in this study. One was the time point when the samples were placed for 0 h when no degradation has occurred. As the target time for prediction, the gene expression data at that time point are most desirable to be known. As a complement, another time point was chosen that was the time point of the sample place for a period of time (6 h). A list of genes associated with RNA degradation in existing studies was found through NCBI. The sampling points represent the expression values of genes corresponding to 12 time points for the 60 genes.

The raw gene expression data are shown in Figure 2a, and the data to be predicted are highlighted with blue dots. Through Figure 2a, it can be seen that the values of the expression of the genes in these time series show a periodic oscillation. In general, these data present a nonlinear, nonsmooth time–frequency signal. Specifically, gene expression data related to the apoptosis and degradation of RNA samples at different time points were utilized, and the gene names and expressions are provided (Appendix A). These prediction points are the gene expression data at 0 h when the RNA has not yet undergone degradation. It can be seen that these data exhibit complex nonlinearity and nonstationarity, which makes the prediction of gene expression at specific locations extremely difficult. How to solve the nonlinearity problem of gene expression data and make the deep learning neural network operate on a more regular time is the key to improve the prediction accuracy and model robustness. In this regard, the empirical mode decomposition (EMD) [16] signal decomposition technique was utilized, which aimed to reduce the nonlinearity of the original gene expression data and decompose them into a series of stable components with different frequencies. This study is based on each component and is combined with a deep learning predictor, which is expected to improve the accuracy of model analysis.

The decomposition results are shown in Figure 2b. After the EMD component, the original gene expression signal is divided into nine different frequency components of intrinsic mode functions (IMFs) [17]. As a new data analysis method, EMD has obvious advantages in dealing with nonsmooth and nonlinear data. EMD is especially suitable for handling terahertz time-domain signals. Compared with the simple harmonic function, IMF does not have a constant amplitude and frequency in the simple harmonic function, but has an amplitude and frequency that vary as a function of time. The system can achieve remarkably less reconstruction errors at extremely low oversampling rates. The different components contain different gene expression information. The correlation between each component and the original data is quantified by Spearman’s [18] rank-order relationship, and the correlation heat map is shown in Figure 2c. It is obvious that different components contain different gene expression information, and aggregating all components will obtain the total gene expression information. This indicates the high reliability of the multi-scale analysis method proposed in this study. Meanwhile, compared with the original data in Figure 2a, each component shows the trend of the recent harmonic function and has a strong change pattern. IMF can not only be amplitude-modulated but also can modulate frequency. The variable instantaneous frequency and instantaneous amplitude can enhance the efficiency of signal decomposition. At the same time, the EMD method can better preserve the nonsmooth and nonlinear characteristics of the original signal. The eigenmode function obtained by EMD also has the characteristic of intra-wave modulation, which can cover the information of the same component expressed by different Fourier frequencies into one IMF [19]. The EMD method is not only intuitive, simple, complete, approximately orthogonal, and modulated in the IMF components, but also has good adaptive properties. The completeness means that the IMF components obtained by the decomposition can fully recover the original signal. The adaptiveness of the EMD method is mainly manifested in the adaptiveness of the generated basis functions, the adaptive frequency resolution of the IMF, and the adaptive filtering characteristics. The adaptive basis function means that the decomposition quantities obtained by EMD for signal decomposition are an adaptive generalized basis from the perspective of signal analysis. A series of IMFs with variable frequency and variable amplitude are obtained by the fully adaptive signal in the decomposition process. Finally, the trends of the data can be effectively captured using LSTM, which provides an accurate prediction of unknown gene expression.

### 2.2. Construction of Core Predictor Model Using LSTM 

First, the LSTM model was built for each IMF component to obtain the corresponding prediction results. The prediction results of the nine components are shown in Figure 3a–i. Then, we performed the predictions of the gene expression according to these nine modules separately. Specifically, the white dots represent the true values and the blue dots represent the predicted values. The line of blue dots (predicted values) in the results will be very close to the line of white dots (true values). From Figure 3a–i, it can be seen that the initial signal is decomposed into nine IMFs by EMD, and the nonsmooth signal becomes increasingly smooth. It can also be seen that the LSTM can accurately capture the change trend of each component after decomposition, and the prediction accuracy is satisfactory. Figure 4a–i show the comparison of the predicted results of each component with the original results. As a result, the prediction values of each component are closer to the original data. There is no difference between the prediction results of each component and the original data, and all the *p*-values > 0.05.

In order to quantify the prediction accuracy of each component more directly, the root-mean-square error (*RMSE*), mean absolute error (*MAE*), and coefficient of determination (*R*^2^) were selected as the prediction performance evaluation indexes. Among them, *RMSE* can be used to assess the degree of inconsistency between the predicted and true values of the model. Theoretically, the smaller its value is, the better the performance of the model. In addition, the *MAE* can be used to complement the evaluation of the performance of this model. Meanwhile, the induced effects of the model can be evaluated by the determinants statistic *R*^2^. The *R*^2^ is an excellent statistical indicator of the closeness between the predicted and true values. The results of *R*^2^ of each component are shown in Figure 5a–i. In general, the performance of this model can be fully evaluated by the above three items. The results of the evaluation of each component are shown in Table 1. These results illustrate the effectiveness of the multi-scale analysis based on EMD in various aspects.

### 2.3. Target Time Point Prediction Based on LSTM 

The final gene expression prediction results can be obtained by recombining the prediction results of each component. We made predictions for the two time points of 0 h (Figure 6a,b) and 6 h (Figure 6c,d) in the original data, respectively. Both predictions were excellent either by predicting the initial unknown time point from the known later time point or by predicting the later unknown time point from the known earlier time point. In addition, we calculated the determinant statistics *R*^2^ between the predicted results and the true values. It can be seen that the fit between the predicted and true values is satisfactory, and more than 90% of the data points are located in the elliptical confidence band of the fitted curve (Figure 6b,d). The forward time prediction results have an *RMSE* of 0.2558, *MAE* of 0.2105, and *R*^2^ of 0.4771. The overall prediction error is low and there is a high agreement between the prediction results and the true values. The *RMSE* value for the reverse time prediction is 0.4002, the *MAE* value is 0.3169, and the *R*^2^ is 0.6145. It also shows low error and excellent agreement between the prediction results and the true values. The prediction results in both directions indicate that this prediction model can obtain well the gene expression results at an unknown destination time. In summary, it can be seen that our proposed Multi-LSTM model can dig deeper into the change pattern of gene expression. Moreover, the prediction results can excellently reflect the expression level of genes at an unknown time.

The model combines signal processing technology and deep learning with biomedicine, and can be widely applied to gene function prediction, disease-related gene discovery, and transcriptomic analysis. In the present study, only the expression data of RNA degradation-related genes were utilized, and the data volume can be expanded in the future to apply Multi-LSTM to a wider range of fields. This model can also be used for time-related disease prediction, for example, degenerative brain diseases and Parkinson’s disease (PD). To use this model for disease prediction in Parkinson’s disease, the genes related to RNA degradation can simply be replaced with genes related to Parkinson’s disease. Of course, this model is not limited to the brain, but can also be used to predict retinal degenerative diseases, such as age-related macular degeneration. In conclusion, this model can be very widely used in disease prediction.

## 3. Materials and Methods

### 3.1. Data Acquisition and Pre-Processing

The data used in this study were obtained from brain tissue samples of normal mice. Both tissues and blood cells should be preserved by proper methods as soon as possible after leaving the optimal cellular life state; otherwise, they will all undergo degradation. This study did not involve biological experiments, and mainly explored the mechanism of RNA degradation in mouse brain tissue. RNA from mouse brain tissue was stored at room temperature (RT) prior to extraction for 3 samples at 0 h time points (labeled 0.1 h, 0.2 h, and 0.3 h), 3 samples at 2 h time points (labeled 2.1 h, 2.2 h, and 2.3 h), 3 samples at 4 h time points (labeled 4.1 h, 4.2 h, and 4.3 h), and 3 samples at 6 h time points (labeled 6.1 h, 6.2 h, and 6.3 h). The data obtained from the final sequencing of the samples were used for the analysis of this study. The 12 time points are 12 consecutive points in time. We used the next 11 time points as the training set and the gene expression data from the very first time point as the prediction set. Then, we found the list of genes associated with RNA degradation in existing studies from NCBI. In total, there are 60 genes corresponding to the expression values from time point 1 to time point 12, respectively. The data used in this study were partly obtained from previously published articles [20]. Time-specific clean data of the build sequencing results have been uploaded to the NCBI database. The alignment of genes from time-series transcriptome sequencing results was performed by Hisat2 [21]. The conversion between data was performed using Samtools [22]. The number of reads mapped to each gene for each sample was calculated using FeatureCounts. A catalog of RNA degradation-related genes was downloaded from NCBI, and the corresponding expressions of the genes associated with each sample were extracted. Data pre-processing includes data cleaning, data integration, data transformation, and the reduction in experimental data before model construction. These processes can improve the training speed of subsequent models and the accuracy and credibility of experimental results. In this study, the raw gene expression data were preprocessed with censoring and normalization. The RNA-seq data were normalized by a heat map using ggplots in the R package. 

### 3.2. EMD-Based Decomposition Signal Processing

EMD has been recognized as an effective time–frequency analysis method for dealing with nonlinear, nonstationary signals since it was proposed in 1998 [23]. This method adaptively decomposes the signal into the sum of several IMFs based on the characteristics of the input signal itself, without any prior knowledge. The method has gradually developed its unique advantages in nonstationary signal processing over the years and has important theoretical research value and broad application prospects. EMD is considered as a major breakthrough from the traditional time–frequency analysis methods such as Fourier analysis and wavelet transform, which are based on linear and smooth assumptions. The EMD method is no longer limited by Heisenberg’s inaccuracy principle and can obtain a high-frequency resolution. The method is based on the decomposition of the signal itself, which does not need to define the basis function in advance, and does not need to adopt the prior knowledge of the signal. Theoretically, each IMF must satisfy the following two conditions. First, on the whole signal, the difference between the number of extreme value points and the number of over-zero points is not greater than 1. Secondly, at any point, the mean value of the upper and lower envelopes is 0. Usually, the actual signals are complex signals and do not satisfy the above conditions. Therefore, Huang et al. [23] made the following assumptions. First, any signal is composed of several eigenmode functions. Second, each eigenmode function can be linear or nonlinear. The number of local zeros and extreme points of each eigenmode function is the same, while the upper and lower envelopes are on the time axis of local symmetry. Third, at any time, a signal can contain a number of eigenmodes. If the modal functions overlap each other, the composite signal is formed.

The variable instantaneous frequency and instantaneous amplitude can enhance the efficiency of signal decomposition. At the same time, the EMD method can better preserve the nonsmooth and nonlinear characteristics of the original signal. The eigenmode function obtained by EMD also has the characteristic of intra-wave modulation, which can cover the information of the same component expressed by different Fourier frequencies into one IMF [19]. The resolution of the IMF is adaptive in the sense that the eigenmode functions obtained by EMD have different characteristic time scales. The instantaneous frequency resolution $f_{i}$ of the ith IMF can be expressed as
(1)$$f_{i}=\frac{f_{i\;max}}{N}$$
$f_{i\;max}$ denotes the highest frequency contained in the ith IMF and N is the number of signal samples. From Equation (1), it can be seen that the frequency resolution of each IMF is different. The frequency resolution of the components containing low-frequency components is high, and the frequency resolution of the components containing high-frequency components is low. The frequency resolution corresponding to each IMF is adaptively obtained. At the same time, there is no relationship with the practical resolution. In this regard, it is completely different from the wavelet analysis in the time and frequency resolution. It is not subject to the constraints of the principle of inaccuracy.

The purpose of the EMD algorithm is to decompose the original signal into some series of IMFs, and then obtain the time–frequency relationship of the signal by the Hilbert transform. Taking the terahertz time-domain spectral signal as an example, the basic calculation process of the EMD algorithm is as follows.

Step 1: The terahertz time-domain spectral signal x(t) is calculated for all the maxima and minima points. All the curves from the upper envelope u(t) are made by connecting all the maxima points with the cubic spline function. The resulting curves from the lower envelope v(t) are made by connecting all the minima points with the cubic spline function.

Step 2: The mean value of the upper and lower envelopes is calculated by the formula:(2)$$m\left(t\right)=\frac{u\left(t\right)+v\left(t\right)}{2}$$

The components of the EMD algorithm are defined as:(3)h(t)=x(t)−m(t)

Step 3: Determine whether the component h(t) satisfies the definition of IMF. If h(t) satisfies the definition of IMF, then h(t) is the first IMF component filtered out, and it is noted as $c_{1}\left(t\right)$. If h(t) does not satisfy the definition of IMF, h(t) is taken as the original data and the above two steps are repeated until the first IMF component is calculated, which is also labeled as $c_{1}\left(t\right)$.

Step 4: The first IMF component is raised from the original signal to obtain the residual signal, which is calculated as follows.
(4)$$r_{1}\left(t\right)=h\left(t\right)-c_{1}\left(t\right)$$

Repeat the above three steps with $r_{1}\left(t\right)$ as the new signal to be analyzed, so that the second IMF component $c_{2}\left(t\right)$ is obtained. Remove the second IMF component from $r_{1}\left(t\right)$ to obtain the new residual term.
(5)$$r_{2}\left(t\right)=r_{1}\left(t\right)-c_{2}\left(t\right)$$

The above steps are repeated continuously until the stopping criterion of the EMD algorithm is satisfied. The stopping criterion of the EMD algorithm uses the Cauchy convergence criterion, a test that requires the normalized squared difference between two adjacent extraction operations to be sufficiently small, as defined by:(6)$$SD_{k}=\frac{\sum_{t=0}^{T}{\left|\begin{array}{cc} \;h_{k-1\;}\left(t\right)\;- & \;h_{k\;}\left(t\right) \end{array}\right|}^{2}}{\sum_{t=0}^{T}\;h_{k-1\;}{\left(t\right)}^{2}}$$

In the formula: T denotes the signal length, and the decomposition ends when $SD_{k}$ is less than the set threshold value.

The root-mean-square error (*RMSE*), also known as root-mean-square deviation, is a commonly used measure of the difference between the values. Its formula is:(7)$$RMSE=\sqrt{\frac{\left(\sum_{i=1}^{n}{\left(y_{i}-{\hat{y}}_{i}\right)}^{2}\right)}{\left(n-1\right)}}$$

The mean absolute error (*MAE*) is the average of the absolute values of the deviations of all individual observations from the arithmetic mean. The mean absolute error avoids the problem of errors canceling each other out and thus accurately reflects the magnitude of the actual prediction error. Its formula is:(8)$$MAE=\frac{\sum_{i=1}^{n}\left|y_{i}-{\hat{y}}_{i}\right|}{n}$$

The coefficient of determination (*R*^2^) is a statistical measure of how close the regression prediction is to the true data point. The formula for $R^{2}$ is:(9)$$R^{2}=\frac{\sum_{i=1}^{n}{\left({\hat{y}}_{i}-\bar{y}\right)}^{2}}{\sum_{i=1}^{n}{\left(y_{i}-\bar{y}\right)}^{2}}$$

In the equations of *RMSE*, *MAE*, and *R*^2^, n is the number of samples and ${\hat{y}}_{i}$ is the expression value of the target gene predicted by the i th sample. $y_{i}$ is the true expression value of the target gene in the ith sample and $\bar{y}$ is the sample mean. 

### 3.3. Construction of Core Predictors Using LSTM

LSTM is a special recurrent neural network (RNN) proposed to solve the deficiency of RNN in the long-range dependency problem [24]. LSTM achieves the selection of new information addition and the control of information accumulation rate by adding a gating mechanism. The addition of gating control makes up for the shortcomings of the RNN network structure. Compared with the classical RNN network in which there is only a single tanh cyclic body, LSTM adds three gate structures and one memory unit. The flow chart is shown in Figure 1b. Let W be the gate weight and b be the bias. The gate structure can be described as:(10)$$\begin{array}{c} g \end{array}\left(x\right)=\sigma \;\left(\begin{array}{c} W_{x\;}+b \end{array}\right)$$

The three gates in the LSTM gate structure are the input gate, the output gate, and the forget gate. The input gate mainly reflects the amount of information stored in the cell state $c_{t}$ at the current moment of the input sequence $x_{t}$ at this time. The output state of the forget gate mainly determines the amount of information from the cell state $c_{t}$ to the output value of the output gate at this moment. Each gate state update of the LSTM is calculated as:(11)$$i_{t}=\sigma \;\left(W_{i}x_{t}+U_{i}h_{t-1}+b_{i}\right)$$
(12)$$f_{t}=\sigma \;\left(W_{f}x_{t}+U_{f}h_{t-1}+b_{f}\right)$$
(13)$$o_{t}=\sigma \;\left(W_{o}x_{t}+U_{o}h_{t-1}+b_{o}\right)$$

Memory unit update:(14)$$\bar{c}{\;}_{t}=\tan h\;\left(W_{c}x_{t}+U_{c}h_{t-1}+b_{c}\right)$$

Status update:(15)$$c_{t}=f_{t}\cdot c_{t}+i_{t}\cdot \bar{c}{\;}_{t}\;$$
(16)$$h_{t}=o_{t}\;\cdot \tan h\left(c_{t}\right)$$

The formula σ is the sigmoid activation function. $W_{i}$, $W_{f}$, $W_{o}$, $b_{i}$, $b_{f}$, and $b_{o}$ are the weight matrices and biases of the input gate, forget gate, and output gate, respectively. $W_{c}$ and $b_{c}$ are the weight matrices and biases of the memory cell after updating, respectively. $U_{i}$, $U_{f}$, $U_{o}$, and $U_{c}$ are the state quantity weight matrices. $h_{t-1}$ is the previous moment state quantity.

## 4. Conclusions

In this study, we propose a method based on signal decomposition techniques and deep learning, called Multi-LSTM. It is a prediction framework to explore the expression changes of RNA degradation-related genes in the initial hours of the sample, which is important for studying the effect of RNA degradation on gene expression levels. The first module decomposes the collected gene expression data by the empirical mode decomposition algorithm to obtain a series of sub-modules with different frequencies in order to improve data stability and reduce modeling complexity. The second module uses an LSTM neural network as the core predictor, aiming to deeply explore the temporal nonlinear relationships embedded in the sub-modules. Finally, the prediction results of sub-modules are reconstructed to obtain the final transcriptome time series gene expression level prediction results. The results show that EMD can efficiently reduce the nonlinearity of the original data, which provides a reliable theoretical support to reduce the complexity and improve the robustness of LSTM models. Meanwhile, the decomposition-combination prediction framework can effectively predict gene expression levels at unknown time points by combining the robust temporal nonlinearity analysis capability of LSTM. The combination of LSTM and the effective EMD decomposition algorithm not only improves the accuracy of prediction results, but also has low prediction errors. The results show that both forward and reverse time series can be predicted accurately. The combination of deep learning neural networks with biomedicine will lead to great breakthroughs in gene function prediction, disease-related gene discovery, and transcriptome time series analysis.

## Figures and Tables

**Figure 1 ijms-23-07532-f001:**
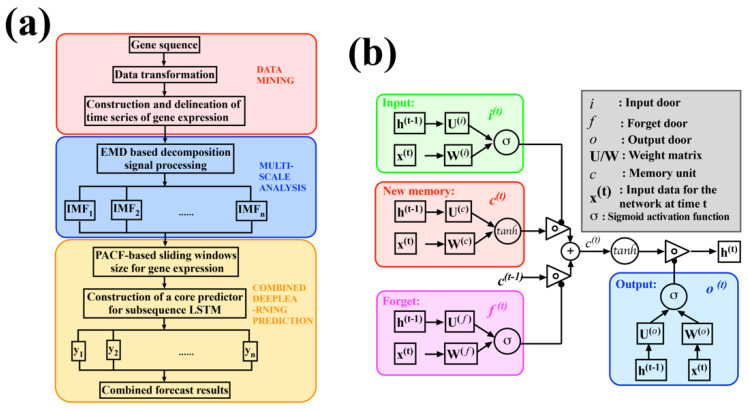
Multi-LSTM flowchart. (**a**): Flowchart of data processing. Step 1: Data mining. RNA degradation-related genes and the expression corresponding to each time are extracted and normalized. Step 2: Multi-scale analysis. The original signal is decomposed into IMF_1_-IMF_9_ by the EMD algorithm, and then the time–frequency relationship of the signal is obtained by the Hilbert transform. Step 3: Deep learning combined prediction. The core predictor is constructed by the LSTM and the prediction results of each subseries are integrated. (**b**): LSTM flow chart. The three gates in the LSTM gate structure are the input gate, the output gate, and the forget gate. The input gate mainly reflects the amount of information stored in the cell state $c_{t}$ at the current moment of the input $x_{t}$. The output state of the forget gate mainly determines the amount of information from the cell state $c_{t}$ to the output value of the output gate at this moment. The output gates of the gate control often use the sigmoid function as the activation function, while the activation functions of the input gates and memory cells usually use tanh.

**Figure 2 ijms-23-07532-f002:**
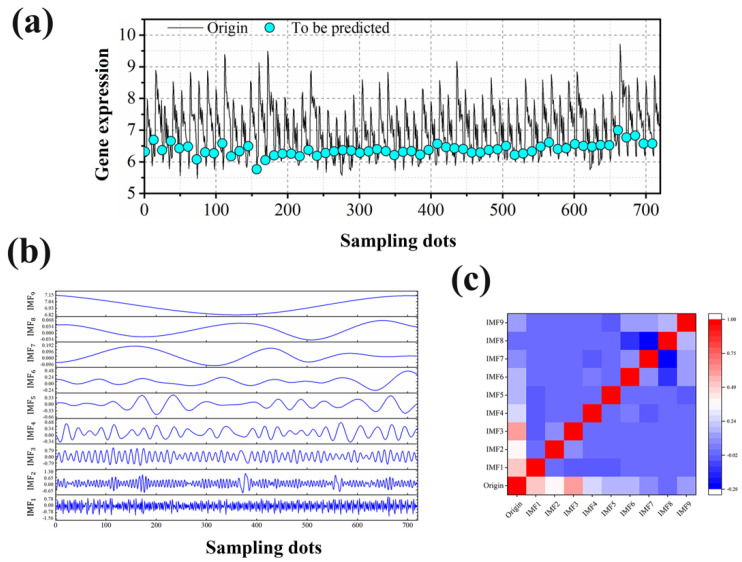
EMD-based decomposition of the original data. (**a**) Expression of the genes to be analyzed. The data to be predicted are highlighted with blue dots. (**b**) The results obtained from the original data based on EMD decomposition for each component of IMF_1_-IMF_9_. (**c**) Spearman correlation of each component of IMF_1_-IMF_9_ after EMD decomposition with the original data.

**Figure 3 ijms-23-07532-f003:**
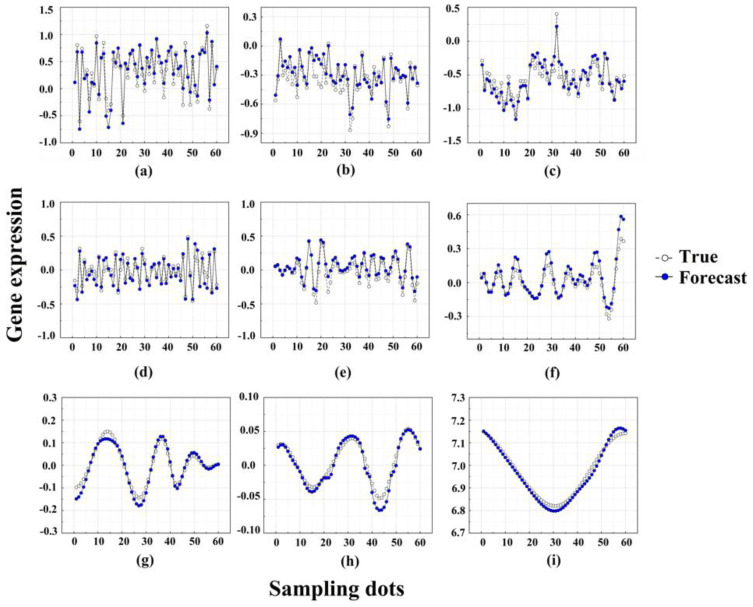
The prediction results of each component using LSTM. (**a**–**i**) correspond to the results of components IMF_1_-IMF_9_, respectively. The expression values of a total of 60 degradation-related genes were decomposed into 9 IMFs by EMD, and the prediction of gene expression was performed according to these 9 modules separately. Specifically, the white dots represent the true values and the blue dots represent the predicted values. From (**a**–**i**), it can be seen that the nonsmooth signal is becoming smoother, and the line of blue dots (predicted values) is also approaching the line of white dots (true values).

**Figure 4 ijms-23-07532-f004:**
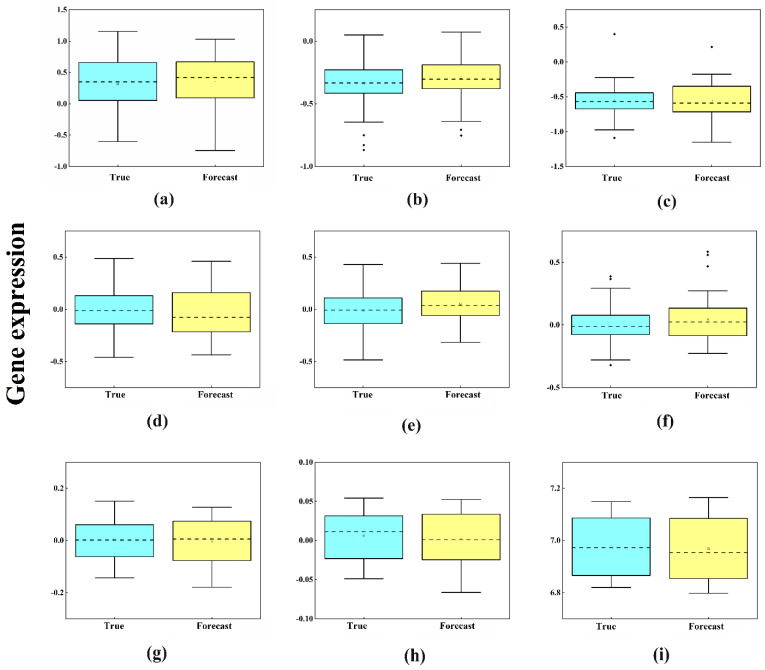
Proximity between the predicted values and the original data. (**a**–**i**) correspond to the results of components IMF_1_-IMF_9_, respectively. The results show that there is no difference between the predicted and true values of each component, and all *p*-values > 0.05.

**Figure 5 ijms-23-07532-f005:**
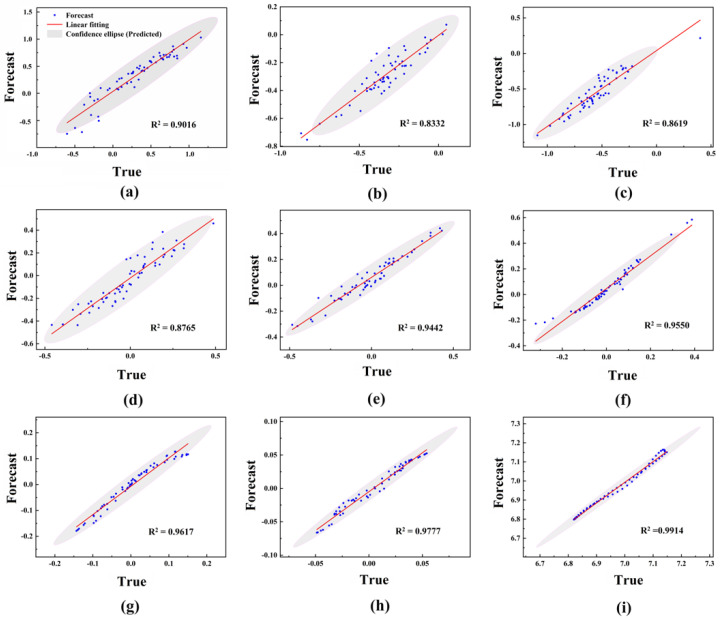
*R*^2^ between predicted and true values of each component. (**a**–**i**) correspond to the results of components IMF1-IMF9, respectively.

**Figure 6 ijms-23-07532-f006:**
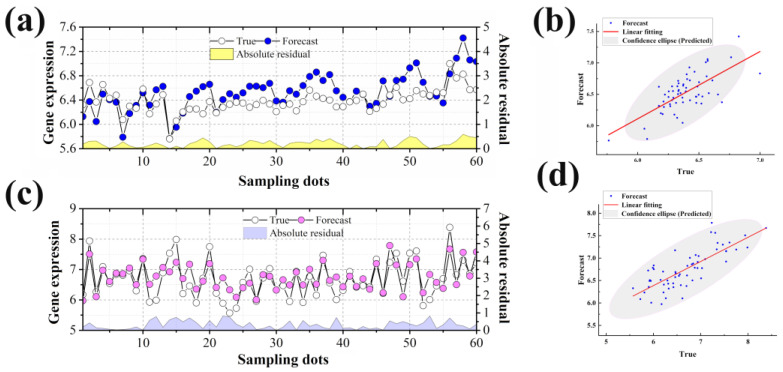
Overall prediction results. (**a**,**b**): *RMSE* value of 0.2558, *MAE* value of 0.2105, and *R*^2^ of 0.4771 at 0 h. The overall prediction error is low and there is a high agreement between the prediction results and the true values. (**c**,**d**): *RMSE* value of 0.4002, *MAE* value of 0.3169, and *R*^2^ of 0.6145 for the reverse time prediction of 6 h. It also shows low error and excellent agreement between the prediction results and the true values.

**Table 1 ijms-23-07532-t001:** *RMSE*, *MAE*, and *R*^2^ values for each IMF component.

	IMF1	IMF2	IMF3	IMF4	IMF5	IMF6	IMF7	IMF8	IMF9
*RMSE*	0.1332	0.0871	0.0959	0.0806	0.0817	0.0688	0.0202	0.0081	0.0172
*MAE*	0.1059	0.0661	0.0812	0.0647	0.0639	0.0492	0.0160	0.0060	0.0151
*R* ^2^	0.9016	0.8332	0.8619	0.8765	0.9442	0.9550	0.9617	0.9777	0.9914

## Data Availability

The data of this study are freely available at https://www.ncbi.nlm.nih.gov/bioproject/PRJNA826142 (accessed on 12 April 2022). The source code used to support the findings of this study is freely available to the academic community at https://github.com/habbyzy/Multi-LSTM (accessed on 23 April 2022).

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
