# Peer review of "Prediction of Time-Series Transcriptomic Gene Expression Based on Long Short-Term Memory with Empirical Mode Decomposition"

_ijms, 2022, doi:10.3390/ijms23147532_

Round 1
Reviewer 1 Report
My suggestions
1. You may mention the sample sources of RNA, and the advantages and disadvantages of different sample types (such as blood, tissue etc.).
2. Some parts of figure 1 and Figure 3 may be difficult to read. Figure 1 may be re-uploaded in a bigger size.
3. Is this model useful in disease prediction? Could you add a few examples of which disease this model could be useful to be predicted?
Author Response
Dear Reviewers:
Thank you for your comments concerning our manuscript entitled “Prediction of Time-Series Transcriptomic Gene Expression Based on Long Short-Term Memory with Empirical Mode Decomposition” (Submission ID: ijms-1793702). Those comments are all valuable and very helpful for revising and improving our paper, as well as the important guiding significance to our research. We have carefully studied these comments and corrected the paper in the hope that it will be more suitable for International journal of molecular sciences.
Responds to the Reviewer :
Comment 1 :
- You may mention the sample sources of RNA, and the advantages and disadvantages of different sample types (such as blood, tissue etc.).
Response:
Thank you very much for your suggestion. RNA degradation always takes place in the tissue that is removed and stored for later processing. However, this process is not only influenced by the time between removing the specimen and the removal of the tissue specimen, but rather by the time between removal of the specimen and the disruption of the blood supply that depends on the type of tissue that is removed.
Both blood and tissues contain RNA, but their RNA content is different. For example, among the various tissue types in mice, the total RNA that can be obtained from brain tissue is approximately 0.80 (ug total RNA/mg tissue). In blood, the total amount of RNA that can be obtained from leukocytes isolated from peripheral blood is 0.50 (ug total RNA/mg tissue). Both tissues and blood cells should be preserved by proper methods as soon as possible after leaving the optimal cellular life state, otherwise they will all undergo degradation. This study did not involve biological experiments, and mainly explored the mechanism of RNA degradation in mouse brain tissue. The main reason for choosing brain tissue is that there are a lot of studies on mouse brain sections, which are very representative. Moreover, our group has been doing research on mouse brain for many years, and the RNA extraction technology of brain tissue is more mature. The sequencing data obtained subsequently will be relatively more reliable.
Thank you again for your suggestion. In the text section, we have also added a sample description section to the method section as follows.
“The data used in this study were obtained from brain tissue samples of normal mice. Both tissues and blood cells should be preserved by proper methods as soon as possible after leaving the optimal cellular life state, otherwise they will all undergo degradation. This study did not involve biological experiments, and mainly explored the mechanism of RNA degradation in mouse brain tissue.”
Comment 2 :
- Some parts of figure 1 and Figure 3 may be difficult to read. Figure 1 may be re-uploaded in a bigger size.
Response:
Thank you very much for your suggestion. We have corrected the size problem of Figure 1 and Figure 3. The latest Figure 1 and Figure 3 are shown below.
Figure 1. Multi-LSTM flowchart. (a): Flowchart of data processing. Step 1: Data mining. RNA degradation-related genes and the expression corresponding to each time were extracted and normalized. Step 2: Multi-scale analysis. The original signal is decomposed into IMF1-IMF9 by EMD algorithm, and then the time-frequency relationship of the signal is obtained by Hilbert transform. Step 3: Deep learning combined prediction. The core predictor is constructed by LSTM and the prediction results of each subseries are integrated. (b): LSTM flow chart. The three gates in the LSTM gate structure are the input gate, the output gate and the forget gate. The input gate mainly reflects the number of information stored in the cell state at the current moment of the input . The output state of the forget gate mainly determines the amount of information from the cell state to the output value of the output gate at this moment. The output gates of gate control often use the sigmoid function as the activation function, while the activation functions of the input gates and memory cells usually use tanh.
Figure 3. The prediction results of each component using LSTM. (a)-(i) correspond to the results of components IMF1-IMF9, respectively. The expression values of total 60 degradation-related genes were decomposed into 9 IMFs by EMD, and the prediction of gene expression was performed according to these 9 modules separately. Specifically, the white dots represent the true values and the blue dots represent the predicted values. From 3a to 3i, it can be seen that the non-smooth signal is getting smoother and smoother, and the line of blue dots (predicted values) are also getting closer and closer to the line of white dots (true values).
Comment 3 :
- Is this model useful in disease prediction? Could you add a few examples of which disease this model could be useful to be predicted?
Response:
Thank you very much for your suggestion. Indeed, making disease predictions is our main purpose of this study. There are lots of clinical diseases are correlated with time. For example, the degenerative brain disease, Parkinson's disease (PD). If you want to use this model for disease prediction for PD, it only need to replace the genes related to RNA degradation with the genes related to PD. There are many Parkinson's disease model mice, and it is possible to extract RNA from the brain of this disease mice for gene prediction at a certain time point. It is even possible to say that this model could be applied to patients with clinical disease, where there has been studies show that there are Parkinson's disease-related target genes in the blood. Therefore, RNA can be extracted and sequenced from blood samples as well, as described in your comment 1. Finally, disease prediction can be performed by this model. Of course, this model is not limited to brain, but can also be used to predict retinal-like degenerative diseases, such as age-related macular degeneration. In conclusion, this model can be very widely used in disease prediction.
Thank you again for your far-sightedness. Your suggestion is a very good direction for development. And, we have added this section to the outlook part. The specific description in the text is as follows.
“This model can also be used for time-related disease prediction, for example, degenerative brain diseases, Parkinson's disease (PD). To use this model for disease prediction in Parkinson's disease, simply replace the genes related to RNA degradation with genes related to Parkinson's disease. Of course, this model is not limited to the brain, but can also be used to predict retinal degenerative diseases, such as age-related macular degeneration. In conclusion, this model can be very widely used in disease prediction.”
Last but not least, thank you very much for your attention and consideration.
On behalf of the authors
Sincerely Yours,
Corresponding author: Qinyu Ge
Mail address: geqinyu@seu.edu.cn

Reviewer 2 Report
In the manuscript, imjs-1793702 by Zhou et al., authors have proposed a method based on signal decomposition technique and deep learning, named Multi-LSTM, which has two modules one decomposes the collected gene expression data by empirical mode decomposition (EMD) algorithm to obtain a series of sub-modules with different frequencies to improve data stability and reduce modeling complexity. The other is based on long short-term memory (LSTM) as the core predictor, aiming to deeply explore the temporal nonlinear relationships embedded in the sub-modules. With the growing number of time-series-based biological experimental data, the time-series information based on this study can be used to achieve in-depth understanding of the gene functions and molecular mechanisms in the biological processes. It is an interesting study, and the data/results are in line with the conclusions. However, general writing style can be improved for better readability.
There are few points to address.
· Line 49-50. “What’s more, it is possible to predict the true gene expression level of RNA when degradation has not occurred by the time of degradation and the existing gene expression level.” Please rephrase the sentence. ‘What’s more’ does not go well.
· Line 62-64. “Artificial intelligence computational methods are particularly important for humans, who can only perform limited experiments directly.” Please replace ‘who’ with ‘which’.
· Line 67, 69 and 71. In such places, please use the ‘First author name et al’ or ‘First authors and colleagues’ or ‘Last author’s group’ to appreciate the articles. For instance, ‘Lakizadeh et al’ or ‘Lakizadeh and colleagues’ instead of ‘Lakizadeh A’.
· In figures, fonts in the axis of graph panels are too tiny to read. Please consider replacing with a better font size or resolution graphs.
Author Response
Dear Reviewers:
Thank you for your comments concerning our manuscript entitled “Prediction of Time-Series Transcriptomic Gene Expression Based on Long Short-Term Memory with Empirical Mode Decomposition” (Submission ID: ijms-1793702). Those comments are all valuable and very helpful for revising and improving our paper, as well as the important guiding significance to our research. We have carefully studied these comments and corrected the paper in the hope that it will be more suitable for International journal of molecular sciences.
Responds to the Reviewer :
Comment 1 :
Line 49-50. “What’s more, it is possible to predict the true gene expression level of RNA when degradation has not occurred by the time of degradation and the existing gene expression level.” Please rephrase the sentence. ‘What’s more’ does not go well.
Response:
Thank you very much for your suggestion. We have changed the sentence to:“If the irreversible change in gene expression level of RNA that occurred during this time can be understood clearly, it will be possible to predict the true gene expression level of RNA when degradation has not occurred by the time of degradation and the existing gene expression level.”Thank you again for your careful review. Your suggestions have helped us to make the context more fluent.
Comment 2 :
Line 62-64. “Artificial intelligence computational methods are particularly important for humans, who can only perform limited experiments directly.” Please replace ‘who’ with ‘which’.
Response:
Thank you very much for your suggestion. We have changed the sentence to:“Artificial intelligence computational methods are particularly important for humans, which can only perform limited experiments directly.” And, we asked our native English-speaking colleagues to read the entire article carefully. Language issues were corrected. Thank you again for your careful review.
Comment 3 :
Line 67, 69 and 71. In such places, please use the ‘First author name et al’ or ‘First authors and colleagues’ or ‘Last author’s group’ to appreciate the articles. For instance, ‘Lakizadeh et al’ or ‘Lakizadeh and colleagues’ instead of ‘Lakizadeh A’.
Response:
Thank you very much for your advice. We have improved all these descriptions in the text. In specific words, the following is shown.“For example, Lakizadeh et al[11] proposed the use of time-series-based gene expression data to complement protein-protein interaction (PPI) networks to detect protein complexes. Wise et al[12] developed SMARTS, a method that combines static and time-series-based data from multiple individuals to reconstruct condition-specific unsupervised feedback networks. Qian et al[13] used deep neural networks for time-series classification tasks.”
Comment 4 :
In figures, fonts in the axis of graph panels are too tiny to read. Please consider replacing with a better font size or resolution graphs.
Response:
Thank you very much for your suggestion. We have corrected the problem with the size of the images. For example, the latest figure 1 and figure 3 are shown below.
Figure 1. Multi-LSTM flowchart. (a): Flowchart of data processing. Step 1: Data mining. RNA degradation-related genes and the expression corresponding to each time were extracted and normalized. Step 2: Multi-scale analysis. The original signal is decomposed into IMF1-IMF9 by EMD algorithm, and then the time-frequency relationship of the signal is obtained by Hilbert transform. Step 3: Deep learning combined prediction. The core predictor is constructed by LSTM and the prediction results of each subseries are integrated. (b): LSTM flow chart. The three gates in the LSTM gate structure are the input gate, the output gate and the forget gate. The input gate mainly reflects the number of information stored in the cell state at the current moment of the input . The output state of the forget gate mainly determines the amount of information from the cell state to the output value of the output gate at this moment. The output gates of gate control often use the sigmoid function as the activation function, while the activation functions of the input gates and memory cells usually use tanh.
Figure 3. The prediction results of each component using LSTM. (a)-(i) correspond to the results of components IMF1-IMF9, respectively. The expression values of total 60 degradation-related genes were decomposed into 9 IMFs by EMD, and the prediction of gene expression was performed ac-cording to these 9 modules separately. Specifically, the white dots represent the true values and the blue dots represent the predicted values. From 3a to 3i, it can be seen that the non-smooth signal is getting smoother and smoother, and the line of blue dots (predicted values) are also getting closer and closer to the line of white dots (true values).
Last but not least, thank you very much for your attention and consideration.
On behalf of the authors
Sincerely Yours,
Corresponding author: Qinyu Ge
Mail address: geqinyu@seu.edu.cn

Round 2
Reviewer 1 Report
The manuscript is acceptable now.